# New Solution for Segmental Assessment of Left Ventricular Wall Thickness, Using Anatomically Accurate and Highly Reproducible Automated Cardiac MRI Software

**DOI:** 10.3390/jimaging11100357

**Published:** 2025-10-11

**Authors:** Balázs Mester, Kristóf Attila Farkas-Sütő, Júlia Magdolna Tardy, Kinga Grebur, Márton Horváth, Flóra Klára Gyulánczi, Hajnalka Vágó, Béla Merkely, Andrea Szűcs

**Affiliations:** Heart and Vascular Centre, Semmelweis University, 1122 Budapest, Hungary

**Keywords:** cardiac imaging, magnetic resonance imaging, cardiomyopathies, myocardium, wall thickness, segmentation, software presentation

## Abstract

**Introduction:** Changes in left ventricular (LV) wall thickness serve as important diagnostic and prognostic indicators in various cardiovascular diseases. To date, no automated software exists for the measurement of myocardial segmental wall thickness in cardiac MRI (CMR), which leads to reliance on manual caliper measurements that carry risks of inaccuracy. **Aims:** This paper aims to present a new automated segmental wall thickness measurement software, OptiLayer, developed to address this issue and to compare it with the conventional manual measurement method. **Methods:** In our pilot study, the algorithm of the OptiLayer software was tested on 50 HEALTHY individuals, and 50 excessively trabeculated noncompaction (LVET) subjects with preserved LV function, whose morphology makes it more challenging to measure left ventricular wall thickness, although often occurring with myocardial thinning. Measurements were performed by two independent investigators who assessed LV wall thicknesses in 16 segments, both manually using the Medis Suite QMass program and automatically with the new OptiLayer method, which enables high-density sampling across the distance between the epicardial and endocardial contours. **Results:** The results showed that the segmental wall thickness measurement values of the OptiLayer algorithm were significantly higher than those of the manual caliper. In comparisons of the HEALTHY and LVET subgroups, OptiLayer measurements demonstrated differences at several points than manual measurements. Between the investigators, manual measurements showed low intraclass correlations (ICC below 0.6 on average), while measurements with OptiLayer gave excellent agreement (ICC above 0.9 in 75% of segments). **Conclusions:** Our study suggests that OptiLayer, a new automated wall thickness measurement software based on high-precision anatomical segmentation, offers a faster, more accurate, and more reproducible alternative to manual measurements.

## 1. Introduction

The thickness of the left ventricular wall varies at different levels and plays a crucial role in different cardiac conditions. Concentric thickening can occur as a physiological adaptation in athletes or as a response to various diseases such as hypertension and aortic stenosis, whereas eccentric thickening serves as an important risk factor for hypertrophic cardiomyopathy (HCM) [1,2,3,4].

Besides thickening, myocardial thinning is also an important prognostic factor, e.g., in dilated (DCM) and arrhythmogenic cardiomyopathy, or in ischemic conditions [5,6,7]. Moreover, new findings provide additional diagnostic insights into compact myocardial thinning in excessive trabeculation associated left ventricular noncompaction (LVET) [8]. However, measuring the compact layer thickness in the presence of an increased trabeculated layer may be challenging.

Dedicated automated or semi-automated analytic software is widely available for volumetric, functional, and muscle mass analysis of CMR and echocardiography, resulting in a fast, reliable and reproducible evaluation [9]. While the accurate measurement of left ventricular wall thickness, especially the compact myocardial layer, may seem deceptively simple, it remains challenging due to technical limitations.

The commonly used manual measurement technique for wall thickness analysis still relies on the use of caliper tools on image series, leading to larger reader-dependent differences [10]. As there are no designated sites in the myocardium where wall thickness measurements should be performed, this results in a lack of both accuracy and reproducibility.

In contrast to the widespread use of contour-based, automated or semi-automated volumetric and functional evaluations, similar compact layer measurement methods based on segmental analysis are still not available in either echocardiography or CMR analytic software. However, there is a clear clinical need for a reproducible, reader-independent method to assess left ventricular wall thickness, since in conditions, such as HCM, DCM or ischemic diseases, accurate thickness measurements are crucial for diagnosis and treatment decisions. Furthermore, myocardial thickness assessment could indicate fibrotic remodeling in ischemic diseases, where contrast agents cannot be applied.

The implementation of artificial intelligence (AI) is gaining site in automatic contour recognition; recent developments in neural networks have led to promising segmentation techniques for MRI in other specialties [11]. Although AI tools continue to struggle with complex myocardial structures such as trabeculae, expert review and complementary methods remain essential to ensure accurate wall thickness measurements.

Therefore, we developed a novel, practical algorithm as part of the used analytic software for segmental wall thickness measurements using contour-based analysis, that can automatically measure a large number of samples within each segment, ensuring high anatomical precision, accuracy and excellent reproducibility.

In this study, we aimed to compare the utility and reproducibility of the conventional manual, and our newly developed semi-automatic wall thickness measurement method used in populations with different myocardial structures, namely healthy individuals and LVET patients, in both sexes.

## 2. Materials and Methods

### 2.1. Study Population

To test the program, we retrospectively selected two representative sample groups from the database of the Heart and Vascular Centre, Semmelweis University. Each subject had previously undergone CMR examination: one group consisted of 50 healthy volunteers (HEALTHY), and the other included 50 subjects with LVET morphology. Both groups consisted of age- and sex-matched Caucasian individuals.

Subjects in the HEALTHY group (average age: 38.6 ± 15.3 years, female: 21) had no known co-morbidities or findings on CMR images.

The LVET group included individuals (average age: 40.1 ± 14.1 years, 20 females) who met the gold standard CMR morphological criteria of LVET, namely, the Petersen (noncompact layer/compact layer distance ratio > 2.3) and Jacquier (noncompact layer/compact layer mass ratio > 2.0) criteria and also had a preserved left ventricular ejection fraction (LV_EF > 50%) [10,12,13].

Subjects were excluded if the short-axis CMR imaging was performed after contrast administration [14]. Individuals with all CMR-incompatible conditions were excluded, such as patients with claustrophobia or arrhythmia, implantable devices in the chest, and poor CMR image quality. All research data were anonymized and only the lead investigator knew the code numbers of the subjects.

The research was designed in accordance with the 1964 Declaration of Helsinki and its amendments, as well as applicable ethical standards. Validated ethical approval of the study was given by the Hungarian Central Ethics Committee (research ethics license number: NNGYK/26676-6/2024), and all participants were fully informed, both verbally and in writing, as acknowledged by their signatures.

### 2.2. MRi Scan

All LVET and HEALTHY participants underwent CMR imaging with 1.5 T scanners (Siemens Magnetom Aera, Siemens Healthineers, Erlangen, Germany; and Philips Achieva, Philips Medical Systems, Eindhoven, The Netherlands) [15,16]. After calibration and axis adjustment sequences, retrospectively gated balanced steady-state free precession (bSSFP) cine sequences were acquired in two-, three-, and four-chamber long-axis views, as well as in contiguous short-axis slices from base to apex, encompassing the entire left ventricle. Images were obtained with an 8 mm slice thickness and no inter-slice gaps. The field of view averaged 350 mm and was adjusted according to body size [17,18]. When contrast agent administration was required, the long- and short-axis images were acquired prior to injection of contrast agent, as pre-contrast images are necessary for accurate assessment of the trabeculated and compact myocardial layers [14].

#### 2.2.1. Measurement of Baseline CMR Parameters

The Medis Suite software (version 4.1; Medis Medical Imaging Systems, Leiden, The Netherlands) with QMass and QStrain modules was used for the postprocessing of the CMR images [19]. In native short-axis images, end-diastolic and end-systolic phases were selected from base to apex, and endocardial and epicardial contours were delineated using automatic tracing, followed by manual adjustments.

For more precise evaluation, a threshold-based algorithm (MassK algorithm 8.1, Medis Suite QMass) was applied with a default threshold setting (50%) [20,21].

This algorithm identifies voxels within the contours as either blood or myocardium based on signal intensity, defining the trabeculated and papillary muscle mass (TPM) within the endocardial, and the total myocardial mass (TM) within the epicardial contour. Compact myocardial mass (CM) was calculated by subtracting TPM from TM (Figure 1). In addition, left ventricular end-diastolic volume (EDV), end-systolic volume (ESV), stroke volume (SV), and ejection fraction (EF) were determined and indexed to body surface area (i). Reference values were based on those reported by Alfakih et al. [22].

#### 2.2.2. Measurement of Wall Thickness

Routine protocols used in CMR evaluations do not include the measurement of the myocardial wall thickness, so this was evaluated separately by two methods.

#### 2.2.3. Conventional Manual Thickness Measurement

For the manual compact layer thickness analysis, long axis images (LAX) in 2-, 3- and 4-chamber views were used. According to current guidelines, wall thickness measurement was based on the American Heart Association (AHA) 17-segment left ventricular model, where thickness of the compact layer was measured in all segments except segment 17, the cardiac apex [23,24]. In this modified 16-segment model, within each segment, one wall thickness measurement was performed using the Medis Suite distance measuring tool (accuracy 2 tenths of a mm). The location of the segment margins and the position of the measured point within segments were determined subjectively by the readers, who were trying to find the average wall thickness of each segment (Figure 2A) [9].

The thickness of the septal and lateral walls in the apical third (Seg. 14 & 16) was obtained by averaging the apical third of the 3- and 4-chamber views.

For a detailed evaluation, we calculated not only the thickness values of each segment, but also the average thickness of the basal (Seg. 1–6), mid (Seg. 7–12) and apical (Seg. 13–16) segments (Figure 2B).

#### 2.2.4. OptiLayer Thickness Measurement

Automated measurements of segmental wall thickness were performed with a newly developed method included in the OptiLayer software (Software version: v1.0; Patent protection registration number; date; facility: P2500327; 1 September 2025; Semmelweis University, Budapest, Hungary). It allows us to determine segmental wall thicknesses using data from Medis Suite analysis.

Firstly, the investigators used the QMass automatic contour detection tool on long-axis images to access the 2-, 3- and 4-chamber endocardial and epicardial contours, with manual corrections where necessary. These contours allow the QMass module to generate a sequence of numbers from automated distance measurements through numerous points of the ventricular wall. Later on, OptiLayer will be able to use this sequence of numbers for segmental calculations.

Subsequently, we developed an annotation method of the apex point which supports the segmentation process, by marking the apex of the heart in all LAX images: at this position readers manually align the endocardial contour with the epicardial contour in an area of approximately 10 mm. With this marking technique, the OptiLayer software can identify the cardiac apex, as the distance between the endocardial and the epicardial contours will be close to zero in the area (Figure 3). This reference point, as an axis, marks the beginning of the segmentation areas, guiding OptiLayer software in the redistribution of the above-mentioned measurement sequences.

Finally, the full distance measurements were exported as CSV files from Medis Suite to the OptiLayer Software.

Using these data, OptiLayer performs anatomical segmentation of the left ventricle based on the modified 16-segment model, by dividing the remaining portion of the myocardial wall between the apex marker and the myocardial base into three equal parts. For thickness evaluation, the program uses an average of 10–12 measurements per segment, and then averages the segmental wall thickness in mm.

The OptiLayer software calculated the average values of each segment and also averaged the wall thickness of the basal (1–6), mid (7–12) and apical (13–16) segments (Figure 2B).

All volumetric, functional and wall thickness analyses were performed by two independent investigators, BM with 5 years of experience and KG with 7 years of experience, to compare results and the accuracy of the method.

Further detail about the calculation method of OptiLayer can be found in Appendix A.

### 2.3. Statistical Analysis

Means with standard deviations (SD) are given as continuous variables, whereas categorical variables are presented as numbers and percentages. To assess distribution normality, we applied the Shapiro–Wilk test. Independent t-tests were used to analyze group differences between LVNC and HEALTHY populations and sexes when data followed a normal distribution, in other cases, Mann–Whitney U tests were performed. Paired t-tests were performed for related parameters with normal distribution and Kruskal–Wallis H-test for related parameters with non-normal distribution.

The reliability of measurements was evaluated using the intraclass correlation coefficient (ICC). Interpretation of ICC values followed standard thresholds: values less than 0.5 were considered to indicate poor reliability; values between 0.5 and 0.75 indicated moderate reliability; values between 0.75 and 0.9 indicated good reliability; and values greater than 0.9 indicated excellent reliability [25]. All analyses were performed using IBM SPSS Statistics (version 30.0), with a significance level set at *p* < 0.05.

## 3. Results

### 3.1. Comparison of Baseline CMR Parameters

Although CMR volumetric and functional parameters were within the normal range in both subgroups, significantly decreased functional and increased volumetric and trabecular muscle mass parameters were observed in LVET subjects, compared to the HEALTHY group. Baseline parameters are shown in Table 1A. All measured parameters showed excellent intraclass correlation (IC) between the two investigators; interobserver variability is shown in Table 1B.

### 3.2. Comparison of Wall Thickness Measurements Using the Manual and the OptiLayer Method

In our study, we compared wall thickness measurements in 16 segments of the LV with the manual and the OptiLayer method in both the total LVET and HEALTHY subgroups.

When analyzing HEALTHY subjects, OptiLayer measured significantly higher wall thickness values in all segments compared to the manual technique (Table 2). For the LVET group, OptiLayer measurements showed greater thickness than manual measurements in nearly all segments, except for the inferior apical (S15) and the anterior and anteroseptal basal segments (S1–S2) (Table 2).

Grouping by sex, among male participants, healthy men exhibited significantly higher wall thickness values in all segments measured with OptiLayer. Similarly, male members of the LVET group showed significantly higher thickness values with OptiLayer in all segments except S1 and S2 (Table 3A).

In contrast, among female participants, healthy women had significantly higher wall thickness with OptiLayer in all segments except the apical segments. Interestingly, female members of the LVET group exhibited significantly higher thickness values with OptiLayer in only four segments (Table 3B).

### 3.3. Comparison of CMR Contours with the New OptiLayer Method

Regarding the manual wall thickness measurement of the LV, we found significantly lower values in the apical segments (S13–16) in the LVET group compared to HEALTHY subjects. In other segments there were no differences between the groups, as shown in Table 4A.

In contrast, OptiLayer measured significantly higher values in HEALTHY subgroup compared to LVET in all but two segments, as shown in Table 4B.

Investigating the IC of the two readers using the manual measurement method, the overall average ICC was 0,66 and five, mainly apical segments had good correlations (>0.75). However, the IC ranges were wide for all measurements, and only one out of 16 segments had the lower IC limit above 0.5 (Table 5A).

When comparing the results obtained with the OptiLayer measurement, excellent (>0.9) or at least good (>0.75) correlations were observed between the two readers in all cases. The correlation ranges fell within the acceptable range with 2 exceptions (Table 5B).

## 4. Discussion

The pathophysiological role of compact layer thickening has long been recognized and the determination of wall thickness has always played an important role in cardiac imaging. One of the most important pathologies in this regard is hypertrophic cardiomyopathy (HCM), where wall thickness is not only a diagnostic tool but also a risk factor [26]. Especially in septal hypertrophy, as it may require therapy because of the LV outflow tract obstruction [27]. Highly eccentric, mainly septal thickening often raises the suspicion of amyloidosis, as many defective proteins may be deposited in this area [28,29,30]. Thickening may also be associated with valvular diseases such as aortic stenosis; furthermore, untreated hypertension is the most frequent cause of concentric thickening [3,4].

Thinning of the left ventricular wall is also a feature of many pathologies; however, it is currently less of a focus than thickening. In coronary artery disease, thinning of the affected myocardium is indicative of chronic myocardial ischemia and is often associated with akinesis or dyskinesis, making myocardial thickness assessment a useful contrast-free marker of fibrotic remodeling [8,31]. Dilated cardiomyopathy (DCM) is also associated with muscle thinning, which may indicate an advanced condition and poor prognosis [7,32]. Recent research has also raised awareness of muscle thinning in LVET, as the morphology is characterized not only by a significantly increased trabeculated noncompact layer but also by a simultaneously decreased compact layer thickness. Furthermore, prognostic effects have been attributed to the wall thickness in LVET [9,33].

Thus, as wall thickness measurements have become increasingly important in the assessment of cardiomyopathies and LVET, we have begun to incorporate it into our CMR evaluation protocol. However, wall thickness measurement highlighted some technical problems that this paper aims to address.

### 4.1. Conventional Measurements of Wall Thickness

According to the current literature, these measurements are performed exclusively using manual caliper tools integrated into standard evaluation software [11]. It is common practice for readers to perform measurements based on the modified 16-segment left ventricular model, which excludes the apex, divides the apical third into four segments and the middle and basal thirds into six segments [24,34]. These areas are defined subjectively by the investigators, based only on what they see as well as on their previous experience; measurements within segments are made roughly in the middle of the segment, with a single measurement.

This manual method of measurement is fundamentally questionable because of its accuracy and reproducibility, and variability between investigators. This was one aspect of our investigation, which used two groups: a HEALTHY and a hypertrabeculated LVET subgroup, in which trabeculae can make accurate wall thickness assessment more difficult. The literature also showed that baseline volumetric muscle mass and functional CMR parameters of LVET were also significantly different from HEALTHY population [35,36].

In terms of comparison of the HEALTHY and the LVET populations using the manual method, segmental wall thickness measurements showed a significant difference in apical segment thickness between the two groups. This difference yielded the expected outcome, as the apical third of the myocardium in individuals with LVET generally has a noticeably thinner myocardium in addition to a pronounced trabeculated layer as a hallmark feature of the morphology [13].

However, when interobserver reliability is evaluated, intraclass correlation coefficients (ICC) fall into the moderate or even poor reliability categories. Furthermore, the confidence intervals associated with these estimates are remarkably wide and frequently fall below 0.5, underscoring the limited reliability of the measurements. When the segmental values are aggregated into apical, midventricular, and basal thirds, this does not reduce this uncertainty.

This shows that manual measurements are inaccurate, as there is no standard point at which wall thickness should be measured within a segment. Obtaining reliable results would require taking many more measurements within a segment and averaging them, which would be an extremely time-consuming solution.

### 4.2. Automated Measurements

Automated wall thickness measurement software may offer a solution to the problems described above. In general, automated or semi-automated image analysis systems have high accuracy due to their large sampling possibilities and also provide very good reproducibility for all users [10]. This was also demonstrated in our study measuring functional, volumetric and muscle mass parameters using the Medis Suite QMASS module with two independent investigators, which provided excellent agreement, due to the semi-automated analysis [35,37].

Regarding the wall thickness measurement, after a thorough examination of the Medis Suite QMASS module, it was found that the module is able to perform automatic distance measurements between the endocardial and the epicardial contours generated semi-automatically on the 3 LAX images. The process of the wall thickness measurements starts from the basal point of the wall, following a counter-clockwise path through the apex to the contralateral basal point, where sampling is performed in 100 locations per plane. However, these measurements are not anatomically segmented by the QMASS module but are simply divided into six regions per plane based on mathematical ratios. It is noteworthy that the segmentation points of the heart should be established on the basal-apical axis of the left ventricle, rather than on a proportional basis. Consequently, the wall thickness measurements generated by Medis Suite QMASS cannot be used for segmental wall thickness estimation, as the aforementioned proportional divisions do not follow the anatomical segments, and the result would be inaccurate.

As other CMR analyzing software does not include specified and anatomically validated segmental wall thickness measurement tools, we created a new measurement solution, the OptiLayer software, which can precisely assess the regional wall thickness with high anatomic accuracy and reproducibility, eliminating these technical errors.

OptiLayer uses the above-mentioned automated epicardial and endocardial contour distance measurements from Medis Suite QMASS LAX images, applying correct segmentation. This is achieved by marking the apex of the left ventricle when generating QMASS contours, which allows OptiLayer to determine the exact axis of the heart. OptiLayer uses the 100 automatic distance measurement values from QMASS, which can be exported from Medis Suite QMASS as a single set of numbers. A schematic image of the OptiLayer method is shown in Figure 4A.

According to this, the simplest way to mark the apex of the heart in this 100-distance measurement sequence is to set the contour distance values close to 0 mm in the area of the apex (segment 17). This can be done by touching the QMASS automated endocardial contour to the epicardial contour in the area corresponding to the apex of the heart, and the QMASS will generate measurements close to 0 mm in this area (roughly 4–8 measurements).

Once the cardiac apex is identified, OptiLayer will divide the remaining apical-basal measuring point values, before and after the cardiac apex, into three equal sections, establishing the anatomically accurate basal, mid and apical segment boundaries. The values in each segment are averaged (roughly 10–13 measurements per segment), resulting accurate segmental wall thickness from a larger number of samples. The user surface is shown in Figure 4B.

Since these measurements rely on automatically measured distances between semi-automated contour lines rather than manual caliper measurements, the probability of measurement errors due to poor image quality or interfering structures, such as increased trabecular mass, is much lower. This is because measuring between contours is much easier than measuring a single point without contour lines.

Recent advances in AI are increasingly applied at different levels in medical imaging [12,38,39]. Cardiac analysis software, like TomTec, uses machine learning models to automate echocardio-graphic measurements like left ventricular volumes, global and segmental strain, and dynamic tracking of heart structures, including 3D image reconstruction. Medis Suite’s Qmass module relies on a deep learning model for automatic contouring; however, users must review and adjust contours as needed, reflecting semi-automated rather than fully autonomous AI use.

This limitation stems from a major challenge faced by these tools: frequent misidentification of the endocardial border, particularly around trabeculae, which often leads to measurement inaccuracies. Our software development specifically aimed to address and eliminate these inaccuracies by focusing on careful assessment of the compact myocardial layer. With sufficient annotated imaging data, future AI integration could further improve OptiLayer analysis.

### 4.3. Usability of the OptiLayer Generated Wall Thickness Measurements

The comparison of wall thicknesses measured with the OptiLayer method between the HEALTHY and LVET groups showed significant differences in almost all segments, in contrast to the manual method. Importantly, this comparison highlights a key methodological finding: measurements performed with OptiLayer showed a strong interobserver correlation between the two investigators, underscoring the reliability of the method.

It is noteworthy that consistently lower thickness values were obtained among LVET subjects, which is not surprising, as not only the trabeculated, but also the compact layer is affected in the development of LVET [13].

When comparing the two different methods, we found wider wall thickness values using the OptiLayer than the manual measurement in both the HEALTHY and LVET groups. However, no consistency could be found in the higher measured values between the two methods. This is likely because the OptiLayer measurements are based on average values from multiple points, whereas manual measurements rely on a single, visually estimated average within the segment. This discrepancy further underscores the inherent limitations and reduced reliability of manual measurements.

When male and female subgroups were measured separately, this was the same for both HEALTHY and LVNC males. Interestingly, a similar difference was observed in the female subgroups; however, it did not reach the level of significance, which may be due to the lower number of cases in females (only 21 subjects).

It is important to note that the software can currently only measure average segmental wall thickness; however, in some cases, such as in HCM, the maximum thickness of an area is also an important prognostic factor [26]. This may be a potential subject for further development of the software.

Finally, these results demonstrate that measurements performed with OptiLayer software are reader-independent and provide a high degree of reproducibility, in addition to being a faster and anatomically more accurate method due to the large number of automated samples.

Risk stratification of LVET is only one potential use of OptiLayer software, but it may also be beneficial for other cardiomyopathies or disorders affecting wall thickness.

## 5. Conclusions

OptiLayer is a novel software developed to generate anatomically accurate LV segmental wall thickness measurements based on the semi-automatic contours of the analytic software. The algorithm relies on automatic distance measurements instead of conventional manual calipers, for which no tools are currently available.

In contrast to manual caliper measurements, our data demonstrate that OptiLayer exhibits excellent reproducibility, as evidenced by robust intraclass correlation coefficients. The software reduces user variability by sampling multiple points per segment, overcoming limitations of single-point manual measurements. The high reliability and accuracy of OptiLayer is supported by the clear and significant differentiation between healthy subjects and pathological morphologies such as LVET, underscoring its potential clinical value. Furthermore, the software provides rapid results displayed in a user-friendly bull’s eye graph, enhancing its practicality and ease of use in everyday clinical practice.

In our current study, OptiLayer is used to calculate average thickness values for testing purposes. This development also included maximal or minimal thickness measurement options to increase its compatibility in different conditions.

In summary, these capabilities position OptiLayer as a valuable tool for scientific research and, with further validation, for clinical assessment of various cardiac pathologies such as LVET and other cardiomyopathies. Additionally, integration of artificial intelligence holds promise to further advance its functionality and expand its clinical utility.

## 6. Limitations

As the OptiLayer software is currently based on the measurement sequence of the Medis Suite Qmass module, OptiLayer cannot be used without the module. Future developments will aim to make it independent of Qmass or to integrate it into the module.

A further limitation is that in the current study, the software only measures the average wall thickness of the segment; however, the algorithm can be further developed to perform minimum or maximum distances.

Currently, this is the first pilot study of the software; thus, it has only been tested in a relatively small number of cases. In future studies, we plan to expand the test population for a more accurate assessment. Similarly, we are planning to extend the testing of the software to multicentric or even multi-regional trials for further refinement.

The software can be used for a more accurate assessment and risk stratification for various pathologies and cardiomyopathies; however, it has not yet been tested in these populations, and further studies are required.

## 7. Patents

The OptiLayer^®^ software (Patent protection registration number, date, facility: P2500327, 1 September 2025; Semmelweis University, Budapest, Hungary).

## Figures and Tables

**Figure 1 jimaging-11-00357-f001:**
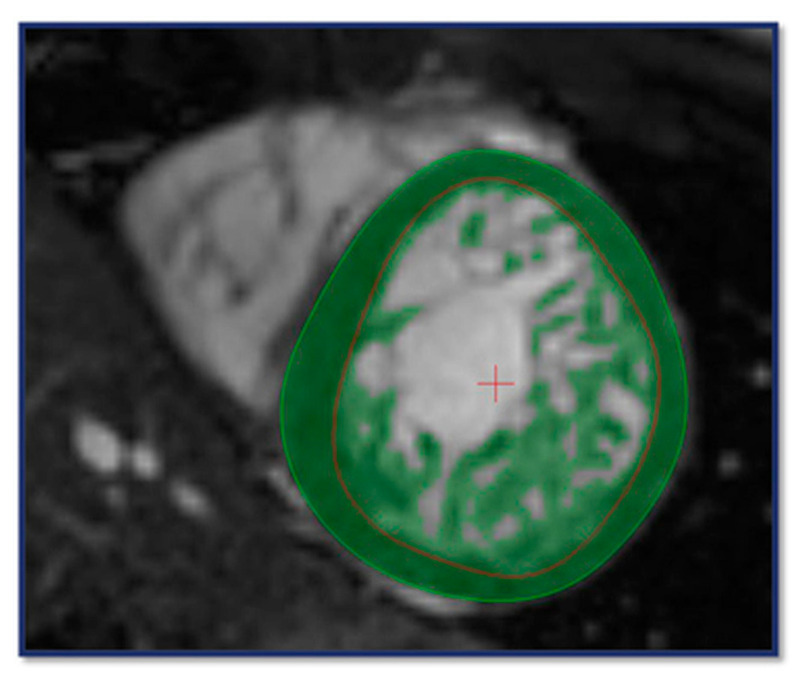
CMR contouring on short axis (SAX) images using the MassK algorithm for the assessment of volumetric, functional and myocardial mass parameters. The light green circle indicates the epicardial-, the red circle refers to the endocardial boundary; the dark green area between the contours represents the compact myocardium. The entire dark green area within the endocardial border highlights the trabeculated and papillary muscle.

**Figure 2 jimaging-11-00357-f002:**
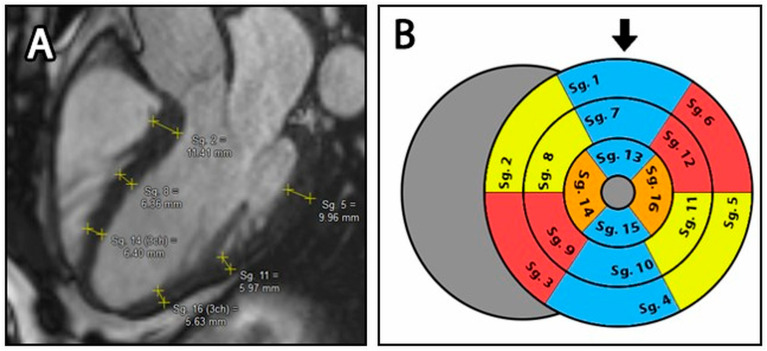
(**A**): Manual distance measurement on the LAX 3 chamber image. Yellow lines represent individually measured wall thickness of each segment. (**B**): Bullseye plot of the LV with the modified 16-segment model. The blue segments represent the 2-cavity plane, the yellow segments represent the 3-cavity plane and the red segments represent the 4-cavity plane. The orange areas indicate regions that belong to both the 3- and 4-cavity areas. *Sg.: Segments, the black arrow represents the anterior wall*.

**Figure 3 jimaging-11-00357-f003:**
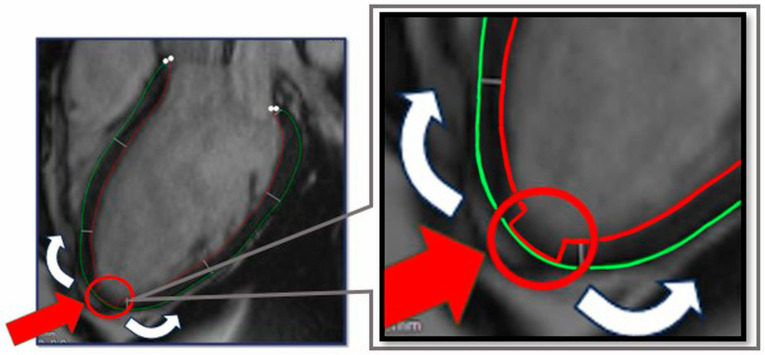
Contour preparation for OptiLayer analysis. The green circle indicates the epicardial-, the red circle refers to the endocardial boundary. The red arrow indicates the aligned epicardial and endocardial contours at the apex of the heart, which represents the origin of the wall thickness analysis from the apical to the basal segments (white arrows).

**Figure 4 jimaging-11-00357-f004:**
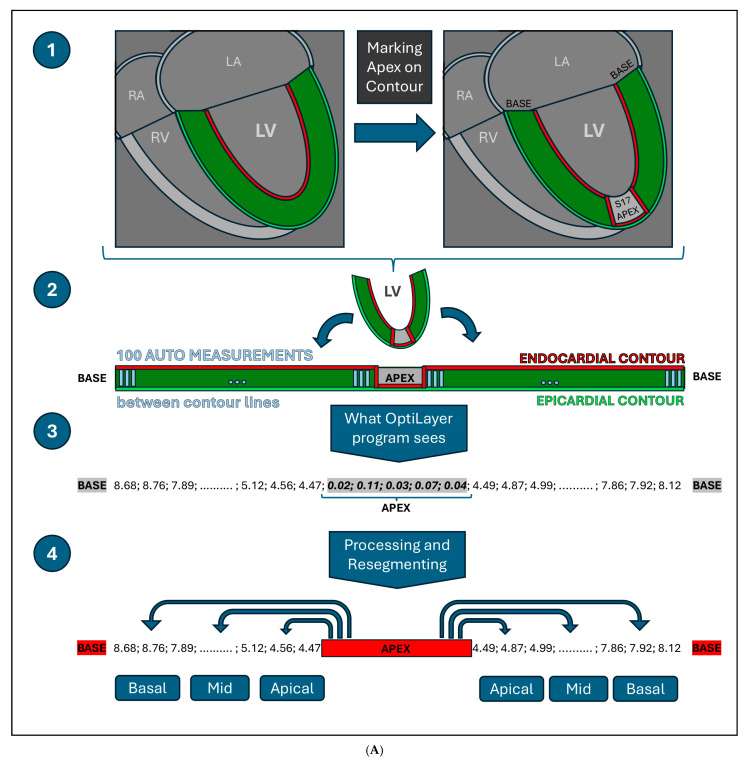
(**A**): Visual representation of the OptiLayer method in schematic chart. The red line indicates the endocardial-, the green line shows the epicardial contours. The dark green area represents the left ventricular wall. *1:* Detection of the cardiac apex; *2*: Measurement direction from the base through the apex to the contralateral base. The gray lines indicate the automated measurement between the epicardial and endocardial contours; *3*: The sequence based on the 100 auto-measured wall distances. Values close to zero indicate the cardiac apex, which is detected by the OptiLayer. *4*: Accurate wall thickness assessment based on anatomical segmentation. The measurement series from the apex to the base is divided into three parts, yielding the wall thicknesses of the basal, mid, and apical segments. *RA: right atrium; LA: left atrium; RV: right ventricle; LV: left ventricle; s17: segment 17; Basal: basal third; Mid: middle third; Apical: apical third.* (**B**): Front end of the OptiLayer program. This screenshot shows the user surface in use, after loading the images, as shown in the left gray panel with test patients. The numbers on the bullseye LV model represent the wall thickness of each segment in mm. Red segments indicate when wall thickness is under 5 mm. *Sg: Segment; A2ch: two chamber view; A3ch: three chamber view; A4ch: four chamber view; 3rd: third; Mid: middle*.

**Table 1 jimaging-11-00357-t001:** Comparison of baseline parameters between study groups, using the automatic MassK algorithm of Medis Suite QMass analytical module (**A**) and assessment of intraclass correlation between readers (**B**).

BASELINE CHARACTERISTICS	(A) FUNCTIONAL PARAMETERS			(B) INTRACLASS CORRELATIONS		
**GROUP**	**LVET**	**HEALTHY**	** *p* **	**ICC**	**Lower Limit**	**Upper Limit**
**LV_EDVI**	73.85 ± 14.84	67.48 ± 10.56	*0.007 #*	*0.981 ***	*0.925 †*	0.995
**LV_ESVI**	26.41 ± 8.21	20.64 ± 4.74	*<0.001 #*	*0.978 ***	*0.919 †*	0.994
**LV_SVI**	47.44 ± 8.93	46.83 ± 7.79	0.359	*0.957 ***	*0.841 †*	0.988
**LV_EF**	64.63 ± 5.78	69.44 ± 4.84	*<0.001 #*	*0.924 ***	*0.719 †*	0.979
**LV_TPMI**	24.81 ± 7.26	19.56 ± 4.64	*0.002 #*	*0.971 ***	*0.903 †*	0.996
**LV_CMI**	48.66 ± 11.05	45.72 ± 7.48	*<0.001 #*	*CD*	*CD*	*CD*
**LV_GCS**	−28.01 ± 4.55	−33.76 ± 5.26	0.238	*0.956 ***	*0.889 †*	0.983
**LV_GLS**	−21.55 ± 2.80	−24.09 ± 3.29	*<0.001 #*	*0.955 ***	*0.889 †*	0.983

*LV: Left Ventricle; LVET: Left ventricular excessive trabeculation; EDV: End-diastolic volume; ESV: End-systolic Volume; SV: Stroke volume; EF: Ejection fraction; TPM: Trabeculated and papillary muscle mass; CM: Compact muscle mass; GCS: Global circumferential strain value; GLS: Global longitudinal strain value; i: BSA indexed parameter; ICC: Intraclass correlation coefficient; p: p value of significance; # p < 0.05; CD: calculated data; ** ICC in excellent range (>0.9); † IC can be accepted according to the sample range (Lower Limit > 0.5).*

**Table 2 jimaging-11-00357-t002:** Comparison of segmental wall thickness measurements using the manual and the new Optilayer method within the total LVET and HEALTHY subgroups.

METHOD COMPARISON						
GROUP (N)	LVET (50)			HEALTHY (50)		
SEGMENT NO.	MANUAL (mm)	OPTILAYER (mm)	*p*	MANUAL (mm)	OPTILAYER (mm)	*p*
**S1**	6.25 ± 1.04	6.53 ± 1.21	0.079	6.63 ± 0.87	8.42 ± 1.36	*<0.001 #*
**S2**	7.05 ± 1.45	7.30 ± 1.84	0.271	7.02 ± 1.05	8.66 ± 1.61	*<0.001 #*
**S3**	6.38 ± 0.87	6.87 ± 1.29	*0.005 #*	6.30 ± 0.71	7.17 ± 1.24	*<0.001 #*
**S4**	6.24 ± 0.77	6.63 ± 1.37	*0.018 #*	6.50 ± 0.82	7.57 ± 1.41	*<0.001 #*
**S5**	6.33 ± 0.94	7.27 ± 1.30	*<0.001 #*	6.45 ± 0.95	7.49 ± 1.09	*<0.001 #*
**S6**	5.98 ± 0.86	6.33 ± 1.13	*0.022 #*	6.00 ± 0.46	7.17 ± 1.08	*<0.001 #*
**S7**	5.33 ± 0.90	6.37 ± 1.60	*<0.001 #*	5.70 ± 0.54	7.02 ± 1.10	*<0.001 #*
**S8**	5.84 ± 0.94	6.54 ± 1.50	*<0.001 #*	5.86 ± 0.58	8.20 ± 1.27	*<0.001 #*
**S9**	6.52 ± 1.02	8.20 ± 2.05	*<0.001 #*	6.62 ± 0.83	8.78 ± 1.68	*<0.001 #*
**S10**	5.71 ± 0.90	6.25 ± 1.30	*<0.001 #*	5.79 ± 0.50	7.73 ± 1.47	*<0.001 #*
**S11**	5.43 ± 0.88	6.74 ± 1.39	*<0.001 #*	5.59 ± 0.41	7.26 ± 1.09	*<0.001 #*
**S12**	5.26 ± 0.86	6.39 ± 1.29	*<0.001 #*	5.67 ± 0.44	7.45 ± 1.28	*<0.001 #*
**S13**	4.60 ± 0.96	5.02 ± 1.10	*0.002 #*	5.31 ± 0.38	5.90 ± 0.89	*<0.001 #*
**S14**	4.79 ± 0.77	5.30 ± 1.07	*<0.001 #*	5.51 ± 0.29	6.56 ± 0.96	*<0.001 #*
**S15**	4.82 ± 0.85	4.93 ± 1.00	0.348	5.52 ± 0.39	6.24 ± 1.04	*<0.001 #*
**S16**	4.80 ± 0.85	5.19 ± 1.07	*<0.001 #*	5.43 ± 0.26	6.21 ± 1.00	*<0.001 #*
**S_AVG_BAS**	6.36 ± 0.69	6.84 ± 1.05	*<0.001 #*	6.48 ± 0.60	7.75 ± 1.05	*<0.001 #*
**S_AVG_MID**	5.68 ± 0.66	6.51 ± 1.16	*<0.001 #*	5.86 ± 0.34	7.54 ± 0.98	*<0.001 #*
**S_AVG_API**	4.75 ± 0.77	5.18 ± 0.96	*<0.001 #*	5.44 ± 0.23	6.28 ± 0.86	*<0.001 #*

*S: Segment; S_avg_BAS: average thickness of the basal third segments (S1-S6); S_avg_MID: average thickness of the middle third segments (S7–S12); S_avg_API: average thickness of the apical third segments (S13–S16); #: p < 0.05.*

**Table 3 jimaging-11-00357-t003:** (**A**,**B**): Comparison of segmental wall thickness measurements using the manual and the Optilayer method in men (**A**) and women (**B**) in LVET and the HEALTHY subgroups.

**(A) METHOD COMPARISON IN MALE (♂)**						
**GROUP (N)**	**HEALTHY ♂ (29)**			**LVET ♂ (29)**		
**SEGMENT NO.**	**MANUAL (mm)**	**OPTILAYER (mm)**	** *p* **	**MANUAL (mm)**	**OPTILAYER (mm)**	** *p* **
**S1**	7.04 ± 0.77	9.18 ± 1.19	*<0.001 #*	6.61 ± 1.15	7.00 ± 1.17	0.103
**S2**	7.61 ± 0.81	9.42 ± 1.28	*<0.001 #*	7.53 ± 1.52	8.03 ± 2.00	0.103
**S3**	6.62 ± 0.63	7.75 ± 1.05	*<0.001 #*	6.63 ± 0.87	7.45 ± 1.17	*<0.001 #*
**S4**	6.83 ± 0.83	8.31 ± 1.30	*<0.001 #*	6.49 ± 0.71	7.17 ± 1.43	*0.004 #*
**S5**	6.93 ± 0.91	8.13 ± 0.82	*<0.001 #*	6.61 ± 0.92	7.75 ± 1.30	*<0.001 #*
**S6**	6.16 ± 0.45	7.76 ± 0.89	*<0.001 #*	6.18 ± 0.89	6.85 ± 1.07	*<0.001 #*
**S7**	5.86 ± 0.54	7.57 ± 1.01	*<0.001 #*	5.50 ± 0.86	7.06 ± 1.63	*<0.001 #*
**S8**	6.15 ± 0.53	8.81 ± 0.89	*<0.001 #*	6.13 ± 0.92	7.15 ± 1.52	*<0.001 #*
**S9**	7.04 ± 0.72	9.79 ± 1.16	*<0.001 #*	6.87 ± 1.07	8.73 ± 2.06	*<0.001 #*
**S10**	5.94 ± 0.52	8.37 ± 1.42	*<0.001 #*	5.93 ± 0.91	6.71 ± 1.29	*<0.001 #*
**S11**	5.74 ± 0.42	7.95 ± 0.70	*<0.001 #*	5.63 ± 0.81	7.21 ± 1.47	*<0.001 #*
**S12**	5.82 ± 0.40	8.08 ± 1.10	*<0.001 #*	5.51 ± 0.84	6.98 ± 1.08	*<0.001 #*
**S13**	5.42 ± 0.41	6.40 ± 0.68	*<0.001 #*	4.87 ± 0.85	5.55 ± 1.02	*<0.001 #*
**S14**	5.63 ± 0.25	7.17 ± 0.70	*<0.001 #*	5.03 ± 0.71	5.76 ± 0.93	*<0.001 #*
**S15**	5.68 ± 0.37	6.85 ± 0.77	*<0.001 #*	5.05 ± 0.69	5.32 ± 1.00	0.079
**S16**	5.55 ± 0.23	6.82 ± 0.77	*<0.001 #*	4.96 ± 0.86	5.65 ± 1.02	*<0.001 #*
**S_AVG_BAS**	6.86 ± 0.44	8.43 ± 0.72	*<0.001 #*	6.67 ± 0.66	7.37 ± 0.98	*<0.001 #*
**S_AVG_MID**	6.09 ± 0.21	8.24 ± 0.56	*<0.001 #*	5.92 ± 0.60	7.10 ± 1.06	*<0.001 #*
**S_AVG_API**	5.57 ± 0.17	6.86 ± 0.58	*<0.001 #*	4.98 ± 0.68	5.64 ± 0.86	*<0.001 #*
**(B) METHOD COMPARISON IN FEMALE (♀)**						
**GROUP (N)**	**HEALTHY ♀ (21)**			**LVET ♀ (21)**		
**SEGMENT No.**	**MANUAL (mm)**	**OPTILAYER (mm)**	** *p* **	**MANUAL (mm)**	**OPTILAYER (mm)**	** *p* **
**S1**	6.07 ± 0.68	7.37 ± 0.74	*<0.001 #*	5.76 ± 0.60	5.88 ± 0.94	0.517
**S2**	6.20 ± 0.75	7.61 ± 1.44	*<0.001 #*	6.40 ± 1.05	6.28 ± 0.88	0.643
**S3**	5.86 ± 0.57	6.37 ± 1.02	*0.015 #*	6.02 ± 0.75	6.06 ± 1.00	0.857
**S4**	6.03 ± 0.55	6.56 ± 0.80	*0.028 #*	5.90 ± 0.71	5.87 ± 0.84	0.884
**S5**	5.78 ± 0.48	6.61 ± 0.75	*<0.001 #*	5.93 ± 0.84	6.61 ± 0.99	*0.004 #*
**S6**	5.77 ± 0.37	6.36 ± 0.73	*0.002 #*	5.70 ± 0.73	5.61 ± 0.77	0.708
**S7**	5.47 ± 0.45	6.25 ± 0.67	*<0.001 #*	5.10 ± 0.93	5.41 ± 0.95	0.176
**S8**	5.45 ± 0.35	6.88 ± 0.78	*<0.001 #*	5.44 ± 0.83	5.70 ± 1.02	0.283
**S9**	6.04 ± 0.59	7.39 ± 1.25	*<0.001 #*	6.04 ± 0.73	7.19 ± 1.61	*0.015 #*
**S10**	5.57 ± 0.40	6.85 ± 1.03	*<0.001 #*	5.39 ± 0.82	5.62 ± 1.06	0.264
**S11**	5.38 ± 0.28	6.29 ± 0.73	*<0.001 #*	5.14 ± 0.91	6.08 ± 0.94	*<0.001 #*
**S12**	5.45 ± 0.42	6.59 ± 1.00	*<0.001 #*	4.91 ± 0.79	5.57 ± 1.11	*0.004 #*
**S13**	5.15 ± 0.27	5.22 ± 0.65	0.616	4.21 ± 0.99	4.28 ± 0.73	0.658
**S14**	5.34 ± 0.25	5.71 ± 0.50	*0.002 #*	4.47 ± 0.74	4.65 ± 0.90	0.689
**S15**	5.30 ± 0.30	5.38 ± 0.71	0.604	4.50 ± 0.96	4.40 ± 0.75	0.617
**S16**	5.26 ± 0.21	5.37 ± 0.58	0.439	4.59 ± 0.83	4.56 ± 0.81	0.759
**S_AVG_BAS**	5.95 ± 0.33	6.81 ± 0.61	*<0.001 #*	5.95 ± 0.50	6.11 ± 0.66	0.245
**S_AVG_MID**	5.55 ± 0.21	6.58 ± 0.46	*<0.001 #*	5.34 ± 0.59	5.70 ± 0.72	*0.009 #*
**S_AVG_API**	5.26 ± 0.18	5.48 ± 0.41	*0.016 #*	4.44 ± 0.79	4.55 ± 0.71	0.391

*S: Segment; S_avg_BAS: average thickness of the basal third segments (S1–S6); S_avg_MID: average thickness of the middle third segments (S7–S12); S_avg_API: average thickness of the apical third segments (S13-S16); #: p <0.05; ♂: Male; ♀: Female.*

**Table 4 jimaging-11-00357-t004:** Comparison of average segmental wall thicknesses between the study groups using manual measurement techniques (**A**) and the new OptiLayer method (**B**).

WALL THICKNESS DIFFERENCES						
METHOD	(A) MANUAL			(B) OPTILAYER		
SEGMENT NO.	LVET (mm)	HEALTHY (mm)	*p*	LVET (mm)	HEALTHY (mm)	*p*
**S1**	6.25 ± 1.04	6.63 ± 0.87	0.026	6.53 ± 1.21	8.42 ± 1.36	*<0.001 #*
**S2**	7.05 ± 1.45	7.02 ± 1.05	0.440	7.30 ± 1.84	8.66 ± 1.61	*<0.001 #*
**S3**	6.38 ± 0.87	6.30 ± 0.71	0.321	6.87 ± 1.29	7.17 ± 1.24	0.114
**S4**	6.24 ± 0.77	6.50 ± 0.82	0.055	6.63 ± 1.37	7.57 ± 1.41	*<0.001 #*
**S5**	6.33 ± 0.94	6.45 ± 0.95	0.263	7.27 ± 1.30	7.49 ± 1.09	0.177
**S6**	5.98 ± 0.86	6.00 ± 0.46	0.448	6.33 ± 1.13	7.17 ± 1.08	*<0.001 #*
**S7**	5.33 ± 0.90	5.70 ± 0.54	0.008	6.37 ± 1.60	7.02 ± 1.10	*0.010 #*
**S8**	5.84 ± 0.94	5.86 ± 0.58	0.454	6.54 ± 1.50	8.20 ± 1.27	*<0.001 #*
**S9**	6.52 ± 1.02	6.62 ± 0.83	0.292	8.20 ± 2.05	8.78 ± 1.68	*0.020 #*
**S10**	5.71 ± 0.90	5.79 ± 0.50	0.288	6.25 ± 1.30	7.73 ± 1.47	*<0.001 #*
**S11**	5.43 ± 0.88	5.59 ± 0.41	0.123	6.74 ± 1.39	7.26 ± 1.09	*0.021 #*
**S12**	5.26 ± 0.86	5.67 ± 0.44	0.002	6.39 ± 1.29	7.45 ± 1.28	*<0.001 #*
**S13**	4.60 ± 0.96	5.31 ± 0.38	*<0.001 #*	5.02 ± 1.10	5.90 ± 0.89	*<0.001 #*
**S14**	4.79 ± 0.77	5.51 ± 0.29	*<0.001 #*	5.30 ± 1.07	6.56 ± 0.96	*<0.001 #*
**S15**	4.82 ± 0.85	5.52 ± 0.39	*<0.001 #*	4.93 ± 1.00	6.24 ± 1.04	*<0.001 #*
**S16**	4.80 ± 0.85	5.43 ± 0.26	*<0.001 #*	5.19 ± 1.07	6.21 ± 1.00	*<0.001 #*
**S_AVG_BAS**	6.36 ± 0.69	6.48 ± 0.60	0.193	6.84 ± 1.05	7.75 ± 1.05	*<0.001 #*
**S_AVG_MID**	5.68 ± 0.66	5.86 ± 0.34	0.038	6.51 ± 1.16	7.54 ± 0.98	*<0.001 #*
**S_AVG_API**	4.75 ± 0.77	5.44 ± 0.23	*<0.001 #*	5.18 ± 0.96	6.28 ± 0.86	*<0.001 #*

*S: Segment; S_avg_BAS: average thickness of the basal third segments (S1–S6); S_avg_MID: average thickness of the middle third segments (S7–S12); S_avg_API: average thickness of the apical third segments (S13-S16); #: p < 0.05.*

**Table 5 jimaging-11-00357-t005:** Agreement of readers between measurements using manual calipers (**A**) and the new OptiLayer method (**B**).

INTRACLASS CORRELATIONS						
METHOD	(A) MANUAL			(B) OPTILAYER		
SEGMENT NO.	ICC	Lower Limit	Upper Limit	ICC	Lower Limit	Upper Limit
**S1**	0.663	0.139	0.902	*0.956 ***	*0.845 †*	0.989
**S2**	*0.799 **	0.382	0.946	*0.981 ***	*0.925 †*	0.995
**S3**	0.487	−0.198	0.844	*0.950 ***	*0.824 †*	0.987
**S4**	0.530	−0.127	0.860	*0.974 ***	*0.901 †*	0.994
**S5**	0.435	−0.266	0.825	*0.909 ***	*0.675 †*	0.977
**S6**	0.565	−0.045	0.870	*0.954 ***	*0.827 †*	0.988
**S7**	0.643	0.081	0.897	*0.894 **	*0.650 †*	0.972
**S8**	*0.770 **	0.319	0.937	*0.948 ***	*0.814 †*	0.987
**S9**	0.525	−0.029	0.850	*0.949 ***	*0.820 †*	0.987
**S10**	0.668	0.082	0.907	*0.920 ***	*0.727 †*	0.979
**S11**	0.524	−0.165	0.859	*0.937 ***	*0.780 †*	0.984
**S12**	0.411	−0.134	0.799	*0.803 **	0.392	0.947
**S13**	*0.913 **	*0.667* *†*	0.997	*0.899 **	*0.667 †*	0.974
**S14**	*0.802 **	0.136	0.954	*0.966 ***	*0.871 †*	0.992
**S15**	0.598	−0.006	0.879	*0.814 **	0.437	0.964
**S16**	*0.792 **	0.380	0.943	*0.940 ***	*0.788 †*	0.985
**S_AVG_BAS**	0.710	0.199	0.919	*0.981 ***	*0.929 †*	0.995
**S_AVG_MID**	0.633	0.023	0.896	*0.958 ***	*0.843 †*	0.989
**S_AVG_API**	*0.837 **	0.447	0.958	*0.938 ***	*0.777 †*	0.984

*S: Segment; S_avg_BAS: average thickness of the basal third segments (S1–S6); S_avg_MID: average thickness of the middle third segments (S7–S12); S_avg_API: average thickness of the apical third segments (S13–S16); ICC: Intraclass Correlation Coefficient; * ICC in good range (>0.75); ** ICC in excellent rage (>0.9); † IC is acceptable according to the sample range (Lower Limit > 0.5).*

## Data Availability

The original data presented in this study are included in the article/Appendix A. Further inquiries can be directed to the corresponding authors.

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
