# Peer review of "New Solution for Segmental Assessment of Left Ventricular Wall Thickness, Using Anatomically Accurate and Highly Reproducible Automated Cardiac MRI Software"

_2313-433X, 2025, doi:10.3390/jimaging11100357_

Round 1
Reviewer 1 Report
Comments and Suggestions for Authors
Regarding the manuscript titled "New solution for segmental assessment of left ventricular wall 2 thickness, using anatomically accurate and highly reproducible 3 automated cardiac MRI software", I would like to inform the authors that this study is innovative. The material presented is well-organized and accurately presented. The quality of the material presented is good. The images are accurate and readable. The method is well presented. The results are also well written. There are only a few minor points that can help improve the quality of this study.
First point: Write the research gap a little more completely in terms of clinical and technological aspects.
Second point: Today, many studies use artificial intelligence for segmentation, which I recommend mentioning artificial intelligence and these studies. Related studies such as the following study can be a suitable study in this area. Also mention the applications of artificial intelligence.
"Enhancing Cancer Zone Diagnosis in MRI Images: A Novel SOM Neural Network Approach with Block Processing in the Presence of Noise"
Third point: The conclusion section is written very limitedly. It would be better to expand this section
The rest of the issues are handled well and I think that by addressing these points the quality of the study will be suitable for publication.
Author Response
Dear Reviewer,
We sincerely thank you for your thoughtful and insightful evaluation of our manuscript. Your questions and comments are of great help and have significantly contributed to enhancing the quality of our work. Below, we address your points in detail.
First point: Could you write the research gap a little more completely in terms of clinical and technological aspects?
Thank you very much for your suggestion. We fully agree that elaborating more on the research gap is important. In response, we have expanded this part in the introduction and discussion sections as well, by including additional clinical perspectives: such as the challenges clinicians face with current segmental assessment methods, and technical aspects like reproducibility and anatomical precision (Lines 68-77, 315-318). This broader framing highlights the significance and novelty of our approach.
Second point: Considering that many studies use artificial intelligence for segmentation, could you mention artificial intelligence and relevant studies, such as "Enhancing Cancer Zone Diagnosis in MRI Images: A Novel SOM Neural Network Approach with Block Processing in the Presence of Noise," as well as the applications of artificial intelligence?
Your awareness of the current trends in segmentation, especially the use of artificial intelligence, is very much appreciated and truly up-to-date. We have incorporated a paragraph on artificial intelligence in the introduction (Lines 73-77), the discussion (Lines 427-439) and the conclusion (Lines 489-490) sections of the manuscript, including the recommended study you kindly suggested. AI can be implemented in different levels to cardiac image recording and analysis. OptiLayer currently utilizes the Medis Suite QMass module, which features deep learning-based automatic contouring. While this automatic contouring tool is recommended to use in our analysis, Medis emphasizes that these contours require user supervision, making it a semi-automated, AI-assisted solution. Furthermore, despite the availability of auto-contouring tools, reliable detection of certain anatomical structures, such as trabeculae, remains challenging. This limitation underscores the importance of expert review and semi-automated methods like ours as essential complements to AI-based tools.
Third point: Would it be possible to expand the conclusion section, as it is currently very limited?
Expanding the conclusion was an excellent recommendation, and we have extended this section to better summarize the clinical and technical implications of our findings and the potential future directions for research in this area (Lines 471-490).
We are very grateful for the time and care you invested in reviewing our work. Your insightful questions and suggestions undoubtedly improved the manuscript, enabling us to present a clearer, well-rounded, and contemporary study. We hope we have clarified all your questions and concerns. Please let us know if you need further details.
Thank you once again for your valuable support and constructive critique.
Kind regards,
Balázs Mester MD & Andrea Szűcs MD PhD
Corresponding Authors

Reviewer 2 Report
Comments and Suggestions for Authors
- “OptiLayer was tested not only in 50 HEALTHY but in 50 excessively trabeculated noncompaction (LVET) subjects with preserved LV function‘ avoid the use of not only but also
- reformulate the abstract in a structured one
- The OptiLayer algorithm measurements were "significantly higher" than those of the manual caliper. The manuscript does not explicitly define a "gold standard" for the true thickness, but the manual measurement is the conventional method for reference. The fact that the new system produces consistently different values couldcause confusion for readers and clinicians. Honestly I cannot get the meaning of the results: If you use the manual segmentation for reference and consistent differences are found between the manual and the automatic measures made by the proposed system, I can supposed that the automatic segmentation you proposed is not reliable. This point is unclear.
- While the authors present this result as a benefit (attributing it to high-density sampling), it means that the values obtained with OptiLayer cannot be directly compared to existing data based on manual measurements without a conversion factor or new reference values
- It seems to me that this system has no effective reliable comparison to assess its accuracy in the current study
- How can you then sustain that this tool is valuable in the clinical parctice?
- Moreover, the OptiLayer demonbstrated greater thickness values in nearly all segments for the LVET group, but there were a few exceptions where the difference was not statistically significant, such as the inferior apical and the anterior and anteroseptal basal segments. This therefore suggests the tool perf ormance may vary depending on the specific anatomical segment and cardiac morphology.
- The "automatic" process still requires some manual steps. Investigators need to manually align the endocardial contour with the epicardial contour at the cardiac apex to create a reference point, needed for the software segmentation. This step could introduce variability, even ifthe authors sustain that the impact is minimal
Author Response
Dear Reviewer,
We sincerely thank you for your detailed and constructive evaluation of our manuscript. Your precise observations and thoughtful critiques have helped us clarify important aspects of our study and highlight areas for further development. Your insights are invaluable in improving the quality and transparency of our work.
REVIEWER COMMENTS AND OUR RESPONSES
Reformulation of the Abstract
Thank you for this suggestion. We have rephrased the methods section of the abstract.
We originally tried to separate the different parts using line breaks for readability. Although the journal’s formatting does not allow visual section separation within the abstract, we included highlighted markers to indicate the distinct sections, as advised (Lines 14-39).
Differences Between OptiLayer and Manual Measurements, and Interpretation of Results
There are divergent opinions in the EACVI CMR Pocket Guide and the 2020 AHA CMR Guideline, on whether the wall thickness should be measured in the short or long axis, as well as on which pathological condition calls for measuring the maximum, minimum, or average wall thickness. For example, in most cases of HCM, the maximum thickness is recommended, whereas in fibrotic DCM, the minimum thickness is also important. In noncompaction, the Petersen recommendations use long-axis (LAX) images to assess thickness. For the sake of method comparability, in this pilot project we aimed to use average wall thickness as the basis without negative or positive peak values; however, the program is also capable of assessing minimum and maximum thickness, allowing flexibility for disease-specific evaluation.
Using our method, the large-scale automated sampling highlights the unreliability of manual average measurements, which are difficult and time-consuming to perform accurately by hand.
With this result, this is precisely the challenge we aimed to address: manual measurements are inherently variable and inconsistent. With possible thickness measurements varying from e.g. 5 to 8 mm within a wall segment, the exact average distance position cannot be determined only with eyeballing. OptiLayer overcomes this by averaging thickness across the entire segment, producing more representative values. This is supported by the intraclass correlation results, as we obtained poor comparability during manual measurements, while good agreements were found using OptiLayer.
As this tool is newly developed, there are no established reference values yet. Our current article introduces the method and demonstrates its advantages while acknowledging the need for further validation in larger, diverse patient populations. The software is accessible for future research, and we have ongoing studies to establish clinical reference standards across various pathologies.
According to the question, we applied modifications in the method section (Line 150), regarding the wall thickness measurements guides, and in the discussion and conclusion section, to improve clarity and potential future clinical use (Lines 471-490).
Variability in Performance Across Cardiac Segments and Morphologies
Our study was a pilot with a modest sample size, which may on one hand explain why some segments such as the inferior apical and selected basal segments did not show statistically significant differences. On the other hand, the apical regions are thinner and more trabeculated in LVET subject, as a signature of the morphology, which makes manual measurement harder to execute correctly, resulting in statistical inaccuracy. Furthermore, due to the thinner compact layer, the variation in the measured values may not be large enough to reach statistical significance between modalities. However, this deficiency is expected to disappear in the clinical phase with a larger number of cases. We further detailed this comparison for clarity in the discussion section. (Lines 449-455)
These findings could suggest anatomy- or morphology-dependence, but further research with larger cohorts is required to clarify segment-specific performance and morphological influences.
Manual Steps Within the “Automatic” Process and Potential Variability
We understand the importance of clarifying what “automatic” entails in our software. In our study, “automatic” refers to the measurement process itself, which does not use calipers manually, only the automatic distance measurement between the contour lines. Yet the analysis does require a manual initialization step: aligning the endocardial contour with the epicardial contour at the apex to establish a reference point. This manual step is necessary but does not directly impact the measurement values and the automatism of distance measurement. This process is also precisely described in the method section of the manuscript. Moreover interobserver variability analyses confirm excellent agreement between measurements, supporting that this manual component does not compromise reliability.
We will clarify this distinction further in the manuscript to avoid any confusion (Lines 78, 83, 369, 403, 432, 471-490).
Thank you for your valuable and constructive review. Your thoughtful comments have helped us elucidate the strengths and current limitations of OptiLayer while outlining a clear roadmap for ongoing development. We greatly appreciate your expert suggestions, which emphasize the importance of transparency and further validation in introducing this promising new tool for cardiac imaging. We hope we have clarified all your questions and concerns. Please let us know if you need further details.
Thank you once again for your valuable support and constructive critique.
Kind regards,
Balázs Mester MD & Andrea Szűcs MD PhD
Corresponding Authors
(This answer is also attached as a separate PDF file)

Round 2
Reviewer 2 Report
Comments and Suggestions for Authors
The authors made the requested changes